# Development and Validation of a Nomogram to Predict Overall Survival in Stage I–III Colorectal Cancer Patients after Radical Resection with Normal Preoperative Serum Carcinoembryonic Antigen

**DOI:** 10.3390/cancers15235643

**Published:** 2023-11-29

**Authors:** Xuan Dai, Haoran Wang, Yaqi Lu, Yan Chen, Yun Liu, Shiyong Huang

**Affiliations:** 1Department of Colorectal and Anal Surgery, Xinhua Hospital, Shanghai Jiao Tong University School of Medicine, Shanghai 200092, China; dx20039@sjtu.edu.cn (X.D.); chen_yan@sjtu.edu.cn (Y.C.); 2The First Clinical School, Xinxiang Medical University, Xinxiang 453003, China; 20201120212@stu.xxmu.edu.cn (H.W.); 20201120220@stu.xxmu.edu.cn (Y.L.)

**Keywords:** colorectal cancer, nomogram, overall survival, CA242, CA125

## Abstract

**Simple Summary:**

A considerable number of patients with colorectal cancer (CRC) do not have elevated serum carcinoembryonic antigen (CEA) levels before undergoing radical surgery for colorectal cancer, but they may still have a poor prognosis. It is imperative to find sensitive and dependable prognostic markers to quickly identify high-risk populations who are susceptible to adverse outcomes in order to offer timely interventions to improve prognosis. In this real-world study of more than 1000 samples, a nomogram based on serum tumor markers and other clinicopathological features was used to accurately forecast the 3-year and 5-year overall survival rates of stage I–III CRC patients after radical resection with normal preoperative CEA, and its predictive power and clinical applicability markedly exceed those of the AJCC 8th TNM stage. In addition, we found the potential prognostic values of carbohydrate antigen 125 (CA125) and carbohydrate antigen 242 (CA242) as supplementary tumor markers in CRC patients who have normal preoperative CEA.

**Abstract:**

We aimed to develop a clinical predictive model for predicting the overall survival (OS) in stage I–III CRC patients after radical resection with normal preoperative CEA. This study included 1082 consecutive patients. They were further divided into a training set (70%) and a validation set (30%). The selection of variables for the model was informed by the Akaike information criterion. After that, the clinical predictive model was constructed, evaluated, and validated. The net reclassification index (NRI) and integrated discrimination improvement (IDI) were employed to compare the models. Age, histologic type, pT stage, pN stage, carbohydrate antigen 242 (CA242), and carbohydrate antigen 125 (CA125) were selected to establish a clinical prediction model for OS. The concordance index (C-index) (0.748 for the training set and 0.702 for the validation set) indicated that the nomogram had good discrimination ability. The decision curve analysis highlighted that the model has superior efficiency in clinical decision-making. NRI and IDI showed that the established nomogram markedly outperformed the TNM stage. The new clinical prediction model was notably superior to the AJCC 8th TNM stage, and it can be used to accurately assess the OS of stage I–III CRC patients undergoing radical resection with normal preoperative CEA.

## 1. Introduction

Colorectal cancer (CRC) stands as a predominant gastrointestinal malignancy, ranking third in global incidence and second in global mortality [1]. In recent years, despite notable advancements in the diagnosis and holistic treatment, the prognosis of CRC has not been significantly improved [2]. Therefore, it is essential to find sensitive and reliable prognostic indicators to identify high-risk groups that are prone to poor prognosis as soon as possible and enact the appropriate interventions to enhance the quality of life and prognosis of patients. The carcinoembryonic antigen (CEA) is a recognized tumor marker that forecasts the prognosis of CRC patients. Patients who have abnormally elevated preoperative CEA levels are inclined to show poor prognosis and recurrence of CRC [3,4,5,6,7]. However, there are still a considerable number of CRC patients whose preoperative serum CEA levels are not elevated [8,9,10], but this does not represent a favorable prognosis for these patients. In fact, some patients have a poor prognosis even if their serum CEA levels are normal [11,12]. For these patients, only monitoring the serum CEA levels has limited benefits, so it is imperative to discover novel, dependable prognostic markers for accurate survival assessments.

In recent years, research prospects have seen a growing interest in the prognostic potential of serum tumor markers other than CEA. A recent study [13] found that elevated preoperative CA125 levels correlated with worse outcomes in patients with metastatic CRC who underwent primary tumor resection (PTR). The carbohydrate antigen 242 (CA242) is a sialic acid-containing carbohydrate antigen that has been reported in relation to the prognosis of CRC [14]. Studies have supported the use of the carbohydrate antigen 19-9 (CA19-9) as a complementary marker for patients with normal preoperative CEA [15], but other studies have suggested that CA19-9 provides limited prognostic information [16]. Carbohydrate antigen 724 (CA724), alpha-fetoprotein (AFP), neuron-specific enolase (NSE), and carbohydrate antigen 211 (CA211) are instrumental in tumor progression, and their clinical value in patients with gastrointestinal cancer has been reported [17,18,19,20,21]. In addition, previous studies [22,23] have shown that the conjoint assessment of multiple serum tumor markers can markedly enhance predictive precision over the use of a singular serum tumor marker. Given the important clinical value of serum tumor markers, it is vital to combine multiple serum tumor markers to achieve a more precise prognostic evaluation.

TNM staging has been widely used for prognostic evaluation. This staging system, which takes into account the depth of invasion, the extent of lymph node metastasis, and so on, serves as a predictive tool for the prognosis of CRC patients. However, other possible prognostic factors, such as age, histologic type, serum tumor markers, and so on, were not considered in this staging system, which may limit the accuracy of prediction to some extent [24,25]. The nomogram is a graphical prediction tool that can intuitively show the probability of endpoint events under the joint action of multiple independent risk factors [26]. Compared with the TNM stage, it is more intuitive and simple to use, and it can also incorporate more risk factors to enhance predictive accuracy. The net reclassification index (NRI) and integrated discrimination improvement (IDI) are the statistical measures used to evaluate the enhanced efficacy of predictive models [27]. By comparing the NRI and IDI values of different prediction models, the efficacy disparities among models can be assessed, allowing for the selection of the optimal predictive model.

In our research, we retrospectively analyzed the association of 15 clinicopathological features, including 7 serum tumor markers, and the prognosis of stage I–III CRC patients after radical resection with normal preoperative CEA. Finally, six variables (age, histologic type, pT stage, pN stage, CA242, and CA125) were chosen to develop the clinical prediction model, and its predictive value was further compared with that of the TNM stage. The results show that the nomogram exhibited a robust predictive capacity. Furthermore, the superiority of the model was further demonstrated by comparing the TNM stage using NRI, IDI and decision curve analysis (DCA).

## 2. Materials and Methods

### 2.1. Research Population

This study encompassed consecutive patients who underwent radical resection of colorectal cancer in the Department of Colorectal and Anal Surgery, Xinhua Hospital Shanghai Jiao Tong University School of Medicine from January 2010 to August 2017 (Figure 1). The exclusions and their criteria are delineated below: (1) 275 patients with distant metastasis; (2) 6 patients without radical resection; (3) 52 patients with pathological non-adenocarcinoma or undetailed pathological data; (4) 156 patients with preoperative neoadjuvant therapy; (5) and 1003 patients without one or more of the eight preoperative tumor markers (CEA, CA242, AFP, NSE, CA125, CA19-9, CA211, and CA724). Finally, our study comprised 1359 participants, of which 277 presented with elevated preoperative CEA, while 1082 exhibited normal preoperative CEA. Patients with normal preoperative CEA were randomized to a training set of 70% (*n* = 758) and a validation set of 30% (*n* = 324). All patients were staged according to the latest NCCN guidelines. All patients included in the study underwent radical (R0) resection of the primary tumor. Adjuvant chemotherapy was administered according to NCCN guidelines to patients who met the criteria for postoperative adjuvant chemotherapy.

### 2.2. The Detection of Tumor Markers

Preoperative tumor markers were measured 1 week before radical surgery for CRC. The threshold values for marker positivity were set as follows: CA242 at 20 U/mL, CEA at 10 ng/mL, AFP at 7 ng/mL, NSE at 16.3 ng/mL, CA125 at 35 U/mL, and CA19-9 at 39 U/mL. For CA211, the cutoff was 3.3 ng/mL, and for CA724, it was 6.9 U/mL.

### 2.3. Follow-Up Study

Patients underwent quarterly follow-ups for the initial two years, biannually for the subsequent 3–5 years, and annually thereafter. Follow-up evaluations included physical examinations, a serum CEA and CA19-9 test, a chest CT scan, and an abdominal and pelvic MRI or CT scan. A colonoscopy was performed every year. The definition of overall survival (OS) was the period from radical resection to either any-cause mortality or the last follow-up. The follow-up evaluation of this study ended on 1 August 2022.

### 2.4. Data Analysis

Categorical variables were compared using the χ^2^ test or Fisher exact test. Continuous variables have been presented as mean ± standard deviation or median (interquartile range (IQR)), and they were compared using the independent sample t-test or the Mann–Whitney U test. Patients included in the study were randomly divided into a training set (70%) and a validation set (30%). The Kaplan–Meier method and log-rank test were employed to evaluate and compare the OS curves of each group. Prognostic factors associated with OS were selected by Cox proportional hazards analysis. Significant variables with *p* < 0.050 in univariate analysis were included under the multivariate Cox regression analysis. We selected variables for inclusion in the nomogram by utilizing stepwise regression according to Akaike’s information criterion (AIC). The probability of 3- and 5-year OS was predicted using the nomogram. Internal validation of the nomogram was conducted based on the validation set. The discrimination ability was evaluated by the concordance index (C-index) and the receiver operating characteristic (ROC) curve. The calibration plot was employed to assess the calibration capacity. NRI and IDI are two mutually complementary validation methods. NRI primarily serves the purpose of comparing the predictive capacity of the old model with the new one. IDI is mainly utilized to examine the overall improvements in the model, assessing its overall performance enhancement. NRI(IDI) > 0—the new model has a better predictive ability compared to the old model; NRI(IDI) < 0—the new model has a poorer predictive ability compared to the old model; and NRI(IDI) = 0—the new model does not provide any predictive improvement compared to the old model. DCA serves as a tool to evaluate a model’s clinical utility, quantifying the net benefit at different threshold probabilities. The curves representing full patient treatment, denoting maximum clinical benefits, and representing no treatment, denoting zero clinical benefit, were employed as benchmarks. Risk stratification was performed based on the median of the risk group scores in the training set to test the differentiation ability of the nomogram.

All statistical tests were performed using SPSS (version 26.0, IBM, New York, NY, USA) and R for Windows (version 4.2.1, http://www.R-project.org/ accessed on 26 November 2023). The R packages used in our study are shown in the Appendix A. All tests were conducted on both sides, with a significance level established at *p* < 0.050.

## 3. Results

### 3.1. Clinicopathologic Characteristics and Comparison of OS Based on Preoperative Tumor Markers

Of the 1359 patients who underwent radical surgery for CRC, 1082 had normal preoperative CEA, while 277 presented with elevated preoperative CEA. The CEA-negative group boasted a 5-year OS rate of 80.04%, markedly surpassing the 59.80% in the CEA-positive group (*p* < 0.0001, Appendix A). The clinicopathological characteristics of patients in CEA-negative and CEA-positive groups are summarized in Appendix A. There were significant differences in the pT stage, pN stage, pTNM stage, and perineural/vascular invasion between the two groups (*p* < 0.050). The median age at operation of CRC patients with normal and elevated preoperative CEA stood at 65 years (interquartile range (IQR): 57–75) and 65 years (IQR: 59–76), respectively. CRC patients with normal preoperative CEA were randomly divided into a training set (758 cases) and a validation set (324 cases). Table 1 presents the detailed clinicopathological characteristics of the participants. The median follow-up of overall CRC patients was 75 months. The 5-year OS event probability for overall CRC patients was 18.7% and the 5-year DFS event probability was 23.6%. In the training set, men (58.6%) accounted for more than women (41.4%), and older patients (51.7%) accounted for more than younger patients (48.3%). Histologic type was adenocarcinoma grade I–II in more than 70% of patients. More than half the patients, 58.3%, presented with a pT stage of T3, while over half, 60.7%, exhibited a pN stage of N0. According to the TNM staging system, 21.6% were categorized as stage I, 39.1% as stage II, and 39.3% as stage III. The majority of patients (91.8%) had no perineural/vascular invasion. The pT stage was earlier in the validation set compared to the training set (*p* = 0.016). Except for the pT stage, all variables displayed no notable disparities between the two sets (*p* > 0.050).

Kaplan–Meier survival analyses of seven serum tumor markers in colorectal cancer patients undergoing radical surgery with normal preoperative CEA showed that, for NSE and AFP, negative and positive groups displayed no notable disparities (*p* > 0.050). The 5-year OS rate of the negative groups of the other five serum tumor markers (CA242, CA125, CA19-9, CA211, and CA724) was better than that of the positive groups (CA242: 81.75% vs. 68.35%; CA125: 80.88% vs. 60.88%; CA19-9: 80.70% vs. 71.78%; CA211: 82.14% vs. 72.25%; CA724: 81.10% vs. 70.22%, *p* < 0.050, Appendix A).

### 3.2. Nomogram Variables Screening

In the univariate Cox proportional hazard regression analysis, eleven variables (age, tumor location, histologic type, pT stage, pN stage, perineural/vascular invasion, CA242, CA125, CA19-9, CA211, and CA724) were correlated with OS (*p* < 0.050, Table 2). Variables that exhibited statistical significance in the univariate analysis were integrated into the multivariate Cox regression analysis. The results show that age (≥65 years vs. <65 years: *p* = 0.001, HR = 1.798, 95% CI: 1.265–2.554), histologic type (adenocarcinoma grade 3 vs. adenocarcinoma grade 1–2: *p* < 0.001, HR = 2.282, 95% CI: 1.483–3.510), pT stage (T3–T4 vs. T1–T2: *p* = 0.012, HR = 1.958, 95%CI: 1.157–3.313), pN stage (N1 vs. N0: *p* = 0.001, HR = 2.102, 95%CI: 1.382–3.197; N2 vs. N0: *p* < 0.001, HR = 5.714, 95% CI: 3.739–8.731), CA242 (positive vs. negative: *p* = 0.013, HR = 1.686, 95% CI: 1.119–2.540), and CA125 (positive vs. negative: *p* = 0.012, HR = 2.133, 95% CI: 1.179–3.859) served as independent determinants for OS (Table 3).

### 3.3. Construction and Validation of Nomogram

The nomogram was constructed based on six variables (age, histologic type, pT stage, pN stage, CA242, and CA125) (Figure 2). For the training set, the C-index stood at 0.748, while for the validation set, it was 0.702. For the training set, the AUC values for the 3-year and 5-year survival of the predictive model stood at 0.796 and 0.765, respectively. Meanwhile, for the validation set, these values were 0.686 and 0.658. These findings underscore the robust predictive efficacy of the model (Figure 3). In addition, the calibration curve showed a strong concordance between the model’s prediction results and the actual observation outcomes (Figure 4).

### 3.4. Comparison of Clinical Value between Nomogram and TNM Stage

To better compare the precision of the nomogram against TNM stage prediction, we analyzed the changes in the C-index, NRI, and IDI. In the training set, the C-index change was 0.080, and the NRI for 3-year and 5-year OS were 0.432 and 0.332, respectively. The IDI values for 3-year and 5-year OS were 0.111 and 0.103, respectively. These findings received further corroboration within the validation set (Table 4). The DCA curves showed that the nomogram precisely forecasted the OS, as the nomogram yielded greater net clinical benefit over a larger range of threshold probabilities than the TNM stage (Figure 5). Overall, this nomogram achieved superior predictive performance and clinical applicability compared to the TNM stage. Participants were grouped into two different risk groups based on the median of the risk group scores in the training set. Kaplan–Meier survival analyses showed better survival for the low-risk group in comparison with the high-risk group, with marked distinctions between them (*p* < 0.010). These were also confirmed in the validation set (Figure 6).

## 4. Discussion

Clinically, CEA is crucial in the prognosis, recurrence and monitoring of CRC [28,29,30]. However, in some clinical trials, more than 50% of CRC patients had preoperative CEA levels within the normal range [31,32,33]. A large European study [10] with 2093 CRC patients discovered that 60.0% of them had normal CEA levels before surgery. CEA was normal in 79.5% of stage I patients, 59.8% of stage II patients, and 52.1% of stage III patients. Similarly, a multicenter study [15] in China focusing on stage II–III CRC patients undergoing radical resection found that nearly 60% of them had normal preoperative CEA levels. In our study, only 20.38% of 1359 CRC patients had preoperative CEA abnormalities, indicating the majority had preoperative CEA values within normal parameters. Yet, the normal value of CEA does not exclude the possibility of poor prognosis [12]. For patients with normal preoperative CEA, there is an urgent need for accurate indicators to predict and evaluate the prognosis. A previous study [34] proposed a CRC-related survival prediction model for preoperative CEA-normal patients, but the number of samples was limited and the predictive significance of serum tumor markers for patients was not fully evaluated, which greatly limited the application of the model in preoperative CEA-normal patients. Therefore, there is a pressing need to devise more precise and holistic clinical prediction models to help clinicians formulate treatment strategies for this subset of patients. This study evaluated the prognostic value of seven tumor markers and other clinicopathological features in patients who had normal preoperative CEA, and constructed a nomogram. The nomogram contains six variables: age, histologic type, pT stage, pN stage, CA242, and CA125. Compared with the TNM stage, this model can provide a more accurate assessment and prediction for patients with normal preoperative CEA.

In our research, we discerned six autonomous factors associated with OS. It is noteworthy that age, pN stage, and pT stage served as prognostic factors influencing OS in the study population. Normal CEA participants with elevated pN and pT stages exhibited a less favorable prognosis, and the prognosis of older patients was worse than that of younger patients, which was also confirmed by Huh et al. [11]. Our findings also indicate that normal CEA patients with “poorly differentiated” pathological grade faced a worse prognosis compared to those graded as “moderately differentiated” or “well differentiated”, which is similar to the findings of Beom et al. [12].

Recently, the assessment of patient prognosis through the inclusion of serum tumor markers has received widespread attention. Paku et al. [35] reported that preoperative high levels of CEA and CA19-9 were related to poor prognosis in patients with locally recurrent rectal cancer. You et al. [36] reported that an increase in the number of positive preoperative serum tumor markers (CEA, CA19-9, CA242, and CA125) was correlated with inferior OS and DFS in patients with stage II and III CRC. Our research findings indicate that, among the seven serum tumor markers, only CA242 and CA125 were independent prognostic factors in participants with normal preoperative CEA, suggesting that they might be complementary markers in patients with normal preoperative CEA. Significantly, previous research [15,37,38] has reported that CA19-9 might serve as a complementary marker to forecast the survival of patients who had normal preoperative CEA, but CA19-9 was not a prognostic factor in multivariate Cox analysis in this study. This means that the preoperative detection of CA242 and CA125 may weaken the impact of CA19-9 on the prognosis of patients with normal preoperative CEA.

CA125 was first used for the detection of ovarian cancer [39], and subsequently, it was also found in gastrointestinal tumor cells [40], where it is vital in tumorigenesis, tumor proliferation, and metastasis [41,42,43]. It has been reported that CA125 is a sensitive predictor of peritoneal metastasis and that it has prognostic value for the recurrence and survival of CRC patients [44,45,46,47]. CA242 is an important tumor marker of digestive tract cancer, and the combination of CEA and CA242 has higher sensitivity than single expression in CRC [6]. Considering the potential predictive value of CA125 and CA242, clinicians should pay greater attention to the supplementary tumor markers CA125 and CA242 when assessing prognosis in patients with normal preoperative CEA.

Although the TNM staging system is important for prognosis, it may not be a comprehensive predictor of patient outcomes [48]. To further improve the accuracy of prediction, clinicians often turn to the use of nomograms in clinical practice. Recently, more and more studies have begun to focus on the clinical application of nomogram. Fu et al. [49] constructed two prognostic nomograms, including CEA and CA19-9, to screen patients with poor prognoses as early as possible. Liu et al. [50] constructed a novel nomogram incorporating CEA, tumor implantation, and number of positive lymph nodes, probably providing an accurate method to evaluate the prognosis of middle-aged and older patients with rectal adenocarcinoma. Similarly, we developed a relatively comprehensive nomogram for stage I–III CRC patients with normal preoperative CEA. In our new model, the pN stage had a significant effect on the total score predicting prognosis, followed by histologic type and CA125. The C-index showed that the model achieved great predictive performance (*p* < 0.050), significantly surpassing that of the TNM stage (*p* < 0.001). The risk score grouping showed that the prediction model had good discrimination ability. In addition, the DCA revealed that our nomogram outperformed the TNM stage in clinical decision-making efficacy. The calibration curves for both groups demonstrated a strong alignment between the predicted and observed outcomes.

NRI and IDI were initially introduced to gauge the enhancement in precision upon the incorporation of new biomarkers into regression models to predict binary outcomes. Recently, these two indices have been extended from binary outcomes to multi-class and survival outcomes [27]. In this research, we used the NRI and IDI to determine the optimal model. NRI and IDI showed that the model exhibited superior accuracy and discriminability in forecasting 3-year and 5-year OS compared to the TNM stage. Therefore, the nomogram we developed may be a useful risk predictor for the assessment of OS in stage I–III CRC patients after radical resection with normal preoperative CEA.

The clinical prediction model we constructed showed the following advantages: At first, the proposed predictive model was applicable to stage I–III CRC patients after radical resection with normal preoperative CEA and could reflect the prognosis of this population accurately. Secondly, the clinical prediction model we constructed was superior to the TNM stage, and it showed better prediction efficiency and clinical decision-making abilities. Thirdly, seven serum tumor markers were incorporated into this research, which fully assessed their predictive value for patients. Finally, serum tumor markers and other clinicopathological features are easily accessible indicators in clinical practice, and a nomogram based on these indicators might hold promising clinical applicability. However, our study has limitations. Firstly, due to the lack of multicenter study data, the nomogram was constructed and validated based on a single database. Therefore, the study is limited by single-institution experience and a lack of external validation. Secondly, the C-index of the nomogram in the validation set was 0.702, which can be used to assess the prognosis of patients with preoperative normal CEA and is smaller than that in the training set. This may be due to the different proportions of pT stage between the training and the validation sets. Thirdly, some factors that may be related to prognosis (such as the BRAF and KRAS mutation status and nutritional status) were not taken into account. Finally, considering the difference in treatment methods between stage I–III patients and stage IV patients, we did not include stage IV patients in the study, which limits the application of this nomogram in stage IV CRC patients.

## 5. Conclusions

Compared with the AJCC 8th TNM stage, our clinical prediction model based on serum tumor markers and other clinicopathological features had more accurate predictive power and better clinical applicability, and it served as a potent tool for forecasting the postoperative overall survival outcomes of stage I–III CRC patients after radical resection with normal preoperative CEA. This research offers a theoretical basis for the detection of other serum tumor markers in CRC patients, such as CA125 and CA242.

## Figures and Tables

**Figure 1 cancers-15-05643-f001:**
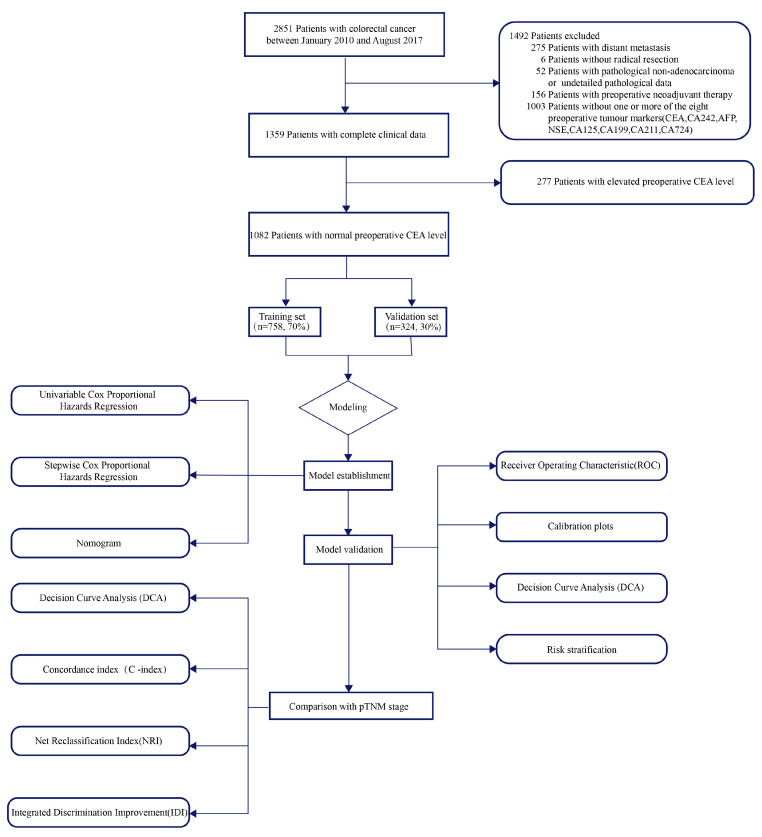
Study flowchart.

**Figure 2 cancers-15-05643-f002:**
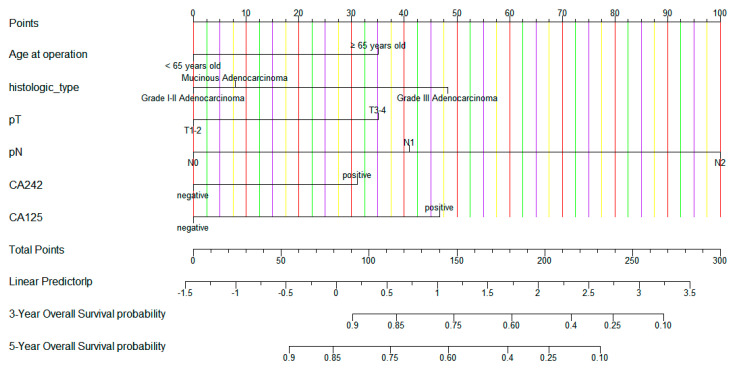
A constructed nomogram for survival prediction of colorectal patients undergoing radical surgery with normal preoperative carcinoembryonic antigen (CEA). For example, a CRC patient after radical resection with normal preoperative CEA was 60 years old at the time of surgery, had a histological type of adenocarcinoma grade III, with pT and pN stages of T3 and N2, respectively, and was negative and positive for CA242 and CA125, respectively. To use the nomogram, each variable’s position on its axis is identified, and lines are drawn from these positions to the points axis. The intersection points determine the number of points attributed to each variable. By summing up the points (230) obtained from all variables, the nomogram provides an estimation of the patient’s likelihood of 3−year and 5−year overall survival (OS). The 3−year and 5−year overall survival rates for this patient were approximately 30% and 12%, respectively.

**Figure 3 cancers-15-05643-f003:**
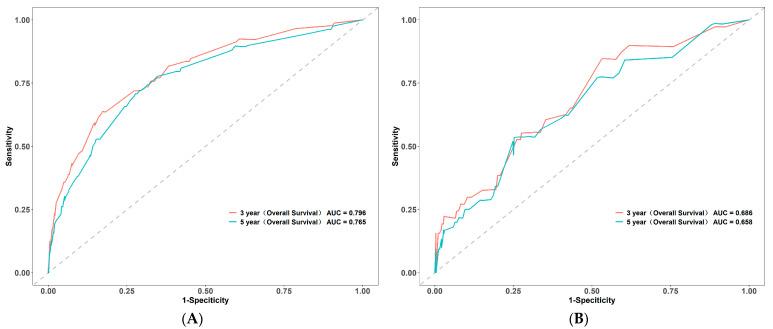
The Receiver Operating Characteristic (ROC) curves for the survival prediction of colorectal cancer patients undergoing radical surgery with normal preoperative carcinoembryonic antigen (CEA) in the training (**A**) and the validation (**B**) sets.

**Figure 4 cancers-15-05643-f004:**
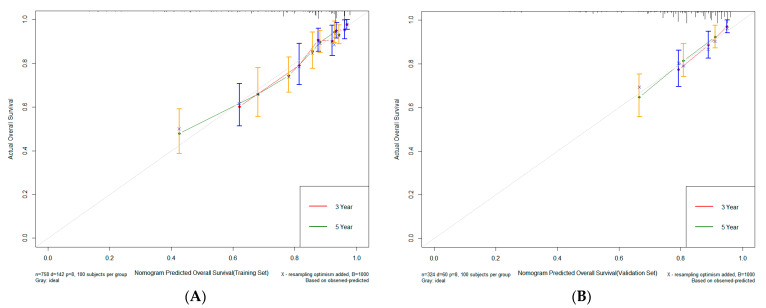
The calibration curves of the nomogram for the survival prediction of colorectal cancer patients undergoing radical surgery with normal preoperative carcinoembryonic antigen (CEA) in the training (**A**) and validation (**B**) sets. The x-axis represents the model’s predicted probability or score, usually in the range of 0 to 1. This is the model’s estimate of the probability of an event occurring. The y-axis represents the actual observed rate of event occurrence (or survival), also in the range 0 to 1. This is the proportion of events that occur in real data. The two curves in red and green represent the calibration curves of the model. The calibration curve shows the relationship between the predicted probabilities of the model and the actual observations. The dotted line represents the perfect calibration line of the theory, which is the 45-degree diagonal line. If the calibration curve coincides with this line, it means that the model’s predictions are perfectly accurate. The error lines (yellow and blue) shown on the calibration curve are used to represent the uncertainty of the calibration curve. These error lines are usually representations of confidence intervals.

**Figure 5 cancers-15-05643-f005:**
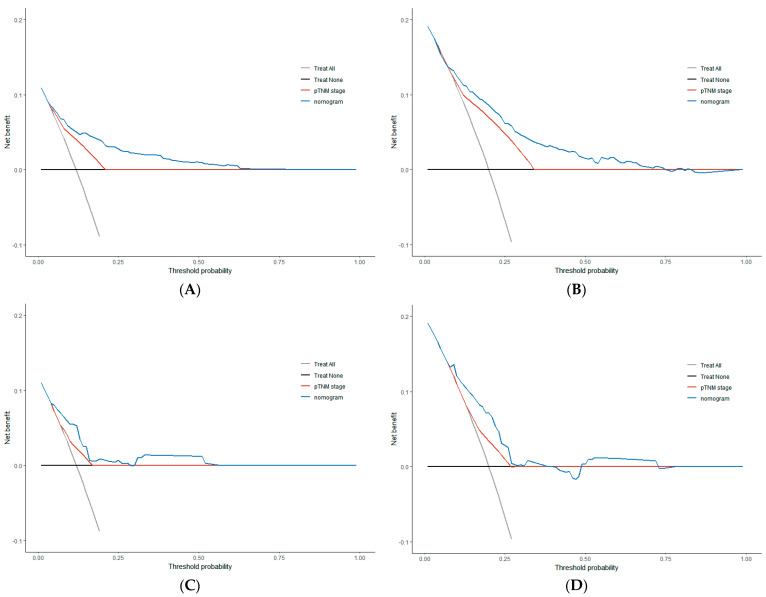
Comparison of decision curve analysis (DCA) results between the nomogram and the AJCC 8th TNM stage for the survival prediction of colorectal cancer patients undergoing radical surgery with normal preoperative carcinoembryonic antigen (CEA). 3−year survival clinical net benefit in the training (**A**) and validation (**C**) sets, 5−year survival clinical net benefits in training (**B**) and validation (**D**) sets.

**Figure 6 cancers-15-05643-f006:**
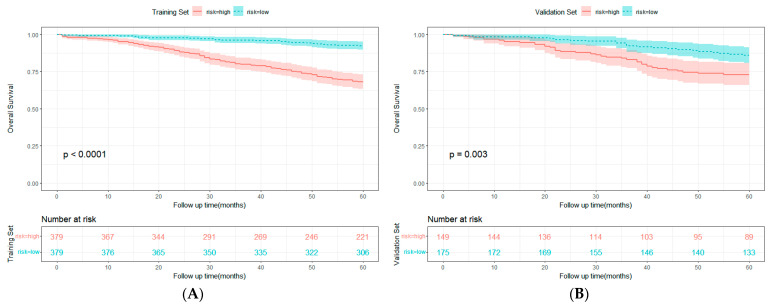
Kaplan–Meier survival curves of colorectal cancer patients undergoing radical surgery with normal preoperative carcinoembryonic antigen (CEA) in the training (**A**) and validation (**B**) sets (the cutoff was stratified by the median of the total score of the training set).

**Table 1 cancers-15-05643-t001:** Comparison of baseline clinicopathologic characteristics between the training set and validation set in colorectal cancer patients undergoing radical surgery with normal preoperative CEA.

Clinicopathological Features	Overall	Training Set	Validation Set	*p* Value
	(*n* = 1082)	(*n* = 758)	(*n* = 324)	
Sex				0.633 ^a^
Male	628 (58.0)	444 (58.6)	184 (56.8)	
Female	454 (42.0)	314 (41.4)	140 (43.2)	
Age (median (IQR))	65 (57, 75)	65 (57, 75)	65.5 (58, 74.2)	0.583 ^b^
Age				0.728 ^a^
<65	518 (47.9)	366 (48.3)	152 (46.9)	
≥65	564 (52.1)	392 (51.7)	172 (53.1)	
Tumor Location				0.749 ^a^
Right Colon	227 (21.0)	158 (20.8)	69 (21.3)	
Left Colon	328 (30.3)	235 (31.0)	93 (28.7)	
Rectum	527 (48.7)	365 (48.2)	162 (50.0)	
Histologic type				0.254 ^a^
Grade I–II Adenocarcinoma	810 (74.9)	566 (74.7)	244 (75.3)	
Grade III Adenocarcinoma	102 (9.4)	78 (10.3)	24 (7.4)	
Mucinous Adenocarcinoma	170 (15.7)	114 (15.0)	56 (17.3)	
pT stage				0.026 ^a^*
T1	82 (7.6)	48 (6.3)	34 (10.5)	
T2	216 (20)	144 (19)	72 (22.2)	
T3	604 (55.8)	442 (58.3)	162 (50)	
T4	180 (16.6)	124 (16.4)	56 (17.3)	
pN stage				0.951 ^a^
N0	654 (60.4)	460 (60.7)	194 (59.9)	
N1	267 (24.7)	185 (24.4)	82 (25.3)	
N2	161 (14.9)	113 (14.9)	48 (14.8)	
pTNM stage				0.180 ^a^
I	248 (22.9)	164 (21.6)	84 (25.9)	
II	406 (37.5)	296 (39.1)	110 (34.0)	
III	428 (39.6)	298 (39.3)	130 (40.1)	
Perineural/Vascular invasion				0.510 ^a^
No	998 (92.2)	696 (91.8)	302 (93.2)	
Yes	84 (7.8)	62 (8.2)	22 (6.8)	
CA242				0.589 ^a^
Negative	936 (86.5)	659 (86.9)	277 (85.5)	
Positive	146 (13.5)	99 (13.1)	47 (14.5)	
AFP				0.704 ^a^
Negative	1061 (98.1)	742 (97.9)	319 (98.5)	
Positive	21 (1.9)	16 (2.1)	5 (1.5)	
NSE				0.428 ^a^
Negative	989 (91.4)	689 (90.9)	300 (92.6)	
Positive	93 (8.6)	69 (9.1)	24 (7.4)	
CA125				1.000 ^a^
Negative	1037 (95.8)	726 (95.8)	311 (96.0)	
Positive	45 (4.2)	32 (4.2)	13 (4.0)	
CA19-9				0.681 ^a^
Negative	999 (92.3)	702 (92.6)	297 (91.7)	
Positive	83 (7.7)	56 (7.4)	27 (8.3)	
CA211				0.483 ^a^
Negative	851 (78.7)	601 (79.3)	250 (77.2)	
Positive	231 (21.3)	157 (20.7)	74 (22.8)	
CA724				0.828 ^a^
Negative	980 (90.6)	688 (90.8)	292 (90.1)	
Positive	102 (9.4)	70 (9.2)	32 (9.9)	
5-year OS				1.000 ^a^
other	880 (81.3)	616 (81.3)	264 (81.5)	
event	202 (18.7)	142 (18.7)	60 (18.5)	
5-year DFS				1.000 ^a^
other	827 (76.4)	579 (76.4)	248 (76.5)	
event	255 (23.6)	179 (23.6)	76 (23.5)	
Follow-up, Median (Q1, Q3)	75 (49, 101)	75.5 (49.2, 101)	75 (41.8, 99.2)	0.596 ^b^

Abbreviations: CA242, carbohydrate antigen 242; AFP, alpha-fetoprotein; NSE, neuron-specific enolase; CA125, carbohydrate antigen 125; CA19-9, carbohydrate antigen 19-9; CA211, carbohydrate antigen 211; CA724, carbohydrate antigen 724. OS, overall survival. DFS, disease-free survival. ^a^ Pearson’s χ^2^ test. ^b^ Mann–Whitney U test. * *p* < 0.05.

**Table 2 cancers-15-05643-t002:** Univariable analyses of clinicopathologic variables in relation to OS in colorectal cancer patients undergoing radical surgery with normal preoperative CEA.

Clinicopathological Features	HR	95%CI		*p* Value
		Lower Limit	Upper Limit	
Sex				
Male	Reference			
Female	0.720	0.509	1.018	0.063
Age				
<65	Reference			
≥65	1.438	1.028	2.012	0.034 *
Tumor Location				
Right Colon	Reference			
Left Colon	0.522	0.328	0.830	0.006 *
Rectum	0.753	0.509	1.115	0.157
Histologic type				
Grade I–II Adenocarcinoma	Reference			
Grade III Adenocarcinoma	3.282	2.170	4.962	0.000 *
Mucinous Adenocarcinoma	1.891	1.238	2.888	0.003 *
pT stage				
T1–T2	Reference			
T3–T4	2.728	1.643	4.527	0.000 *
pN stage				
N0	Reference			
N1	2.514	1.670	3.784	0.000 *
N2	6.121	4.104	9.129	0.000 *
Perineural/Vascular invasion				
No	Reference			
Yes	2.707	1.758	4.170	0.000 *
CA242				
Negative	Reference			
Positive	2.109	1.409	3.156	0.000 *
AFP				
Negative	Reference			
Positive	0.639	0.158	2.579	0.529
NSE				
Negative	Reference			
Positive	1.234	0.722	2.107	0.442
CA125				
Negative	Reference			
Positive	2.707	1.531	4.789	0.001 *
CA19-9				
Negative	Reference			
Positive	1.886	1.121	3.174	0.017 *
CA211				
Negative	Reference			
Positive	1.636	1.135	2.358	0.008 *
CA724				
Negative	Reference			
Positive	1.808	1.127	2.902	0.014 *

Abbreviations: CA242, carbohydrate antigen 242; AFP, alpha-fetoprotein; NSE, neuron-specific enolase; CA125, carbohydrate antigen 125; CA19-9, carbohydrate antigen 19-9; CA211, carbohydrate antigen 211; CA724, carbohydrate antigen 724; OS, overall survival. * *p* < 0.05.

**Table 3 cancers-15-05643-t003:** Multivariable analyses of clinicopathologic variables in relation to OS in colorectal cancer patients undergoing radical surgery with normal preoperative CEA.

Clinicopathological Features	HR	95%CI		*p* Value
		Lower Limit	Upper Limit	
Age				
<65	Reference			
≥65	1.798	1.265	2.554	0.001 *
Tumor Location				
Right Colon	Reference			
Left Colon	0.652	0.402	1.057	0.083
Rectum	1.081	0.715	1.636	0.711
Histologic type				
Grade I–II Adenocarcinoma	Reference			
Grade III Adenocarcinoma	2.282	1.483	3.510	0.000 *
Mucinous Adenocarcinoma	1.128	0.716	1.778	0.602
pT stage				
T1–T2	Reference			
T3–T4	1.958	1.157	3.313	0.012 *
pN stage				
N0	Reference			
N1	2.102	1.382	3.197	0.001 *
N2	5.714	3.739	8.731	0.000 *
CA242				
Negative	Reference			
Positive	1.686	1.119	2.540	0.013 *
CA125				
Negative	Reference			
Positive	2.133	1.179	3.859	0.012 *

Abbreviations: CA242, carbohydrate antigen 242; CA125, carbohydrate antigen 125; OS, overall survival. * *p* < 0.05.

**Table 4 cancers-15-05643-t004:** Comparison between nomogram and pTNM stage in terms of C-index, NRI, and IDI.

Index	Training Set			Validation Set		
	Estimate	95%CI	*p* Value	Estimate	95%CI	*p* Value
NRI (vs. pTNM stage)	0					
For 3-year OS	0.432	0.251–0.601		0.375	0.193–0.572	
For 5-year OS	0.332	0.192–0.481		0.442	0.241–0.655	
IDI (vs. pTNM stage)						
For 3-year OS	0.111	0.061–0.197	0.000 *	0.076	0.030–0.231	0.000 *
For 5-year OS	0.103	0.062–0.162	0.000 *	0.068	0.035–0.180	0.000 *
C-index (OS						
The nomogram)	0.748	0.706–0.791		0.702	0.643–0.761	
The pTNM stage	0.668	0.629–0.706		0.593	0.528–0.658	
Change	0.080	0.051–0.094	0.000 *	0.109	0.040–0.199	0.007 *

Abbreviations: OS, overall survival. * *p* < 0.05.

## Data Availability

The data presented in this study are available on reasonable request from the corresponding author. The data are not publicly available due to privacy.

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
