# Peer review of "Development and Validation of a Nomogram to Predict Overall Survival in Stage I–III Colorectal Cancer Patients after Radical Resection with Normal Preoperative Serum Carcinoembryonic Antigen"

_cancers, 2023, doi:10.3390/cancers15235643_

Round 1

Reviewer 1 Report

Comments and Suggestions for Authors

This article aims to develop and validate a predictive score for survival of cancers with normal CEA.

It carries out a theoretical and statistical analysis of patient data and biological markers in order to define a predictive nomogram for survival and compare it to the score obtained by the TNM classification.

The problem is inherent in the question asked. Indeed, the TNM score is a classification system for solid cancer tumors from the International Union Against Cancer (UICC). It is the most frequently used classification system in oncology. Using this type of classification allows doctors to have a universal language to describe cancer. Its purpose makes it possible to determine the stage of a cancer, and possibly to define a therapeutic strategy. It is not intended to be used as a predictor of survival.

The TNM score is used in this paper to be a reference for estimation of expected survival. In the article, the authors claim to obtain greater precision in the assessment of survival, but their study remains theoretical and must be confronted with the reality of survival, which cannot be evaluated on a cohort of 1082 patients, the precision of their score based on a strictly theoretical analysis of probability.

Finally, they validate their method by a new statistical analysis using a concordance score (C index) to validate their IDI and NRI method. They then allow themselves to conclude that their score has better predictive power than the TNM score... which is not the point!

Furthermore, a non-negligible bias of this type of study is not to take into account the impossibility of defining the importance of each factor T, N or M which could vary from one to the other at each stage, and that ACE has never been defined as a correlating factor for these stages. Each stage T, N or M should be analyzed independently, which is not possible with a cohort of 1000 patients.

Finally, the clinical interest of this nomogram score is not analyzed. Indeed, it does not specify the clinical benefit of the nomogram. It does not specify whether it should have an impact on therapeutic care. It has no real interest in patient care, nor in the statistical reclassification of cancers, the subject of the TNM score.

Comments on the Quality of English Language

Difficult to understand

Author Response

1. Summary

Thank you for your comments and advice. Those comments are all valuable and very helpful for revising and improving our paper. We have studied comments carefully and have made correction which we hope meet with approval. Meanwhile, The manuscript have be revised and edited by English language services of MDPI( The proof can be found below). Based on your instructions and comments, we uploaded the file of the revised manuscript. Revised sections are marked in red in the paper. The corrections in the paper and the responses to your comments are as following:

(Revised manuscript can be found in attachment “revised manuscript”and revised manuscript-supplementary can be found in attachment “revised manuscript-supplementary”)

2. Point-by-point response to Comments and Suggestions for Authors

Comments 1: The problem is inherent in the question asked. Indeed, the TNM score is a classification system for solid cancer tumors from the International Union Against Cancer (UICC). It is the most frequently used classification system in oncology. Using this type of classification allows doctors to have a universal language to describe cancer. Its purpose makes it possible to determine the stage of a cancer, and possibly to define a therapeutic strategy. It is not intended to be used as a predictor of survival.

Response 1:

Thank you for your comments and interest in our study. First of all, it is undeniable that the TNM staging system is the most commonly used international staging system for solid tumours, which allows clinicians to use a common language to describe cancer progression, as well as to assess a patient's cancer stage and develop appropriate treatment strategies. However, in addition to this, TNM staging can also be used for prognostic evaluation. A study[1] published in Ca-a Cancer Journal For clinical showed that the 5-year survival rate of patients with stage I CRC was higher than 90%, while the five-year survival rate of patients with stage IV was only 11% ~ 15%. Cheng et al[2] reported that the overall 5-year survival rate of patients with early-onset colorectal cancer (CRC diagnosed at age <50 years) was 63.2%, with a 5-year survival rate of 93.7% for stage I, 85.0% for stage II, 72.0% for stage III, and 18.7% for stage IV. All of the above suggests that the TNM stage is closely related to patients' prognosis and survival. Konishi et al [3] reported that TNM stage was an independent prognostic factor for recurrence-free survival in patients undergoing radical surgery for stage I-III colorectal cancer and that the risk of patients facing a poor prognosis increased progressively with the higher TNM stage. In addition to colorectal cancer, TNM staging has also been used to assess the prognosis of other cancers[4-6]. Wang et al. [7] constructed and validated a novel model by studying a large cohort of small-cell lung cancer patients from the National Cancer Database (NCDB), and compared it with the eighth edition of the TNM staging, which showed that the model was able to predict the prognosis of the patients more accurately and significantly better than the TNM staging. The results showed that the model was able to predict patients' prognosis more accurately and with significantly better efficacy than the TNM staging. Therefore, the TNM staging system is not only useful for describing cancer progression and formulating treatment plans but also provides an important reference for the prognostic assessment of patients, which can help doctors and patients better understand the progression of the disease and the prospect of treatment.

Comments 2: The TNM score is used in this paper to be a reference for estimation of expected survival. In the article, the authors claim to obtain greater precision in the assessment of survival, but their study remains theoretical and must be confronted with the reality of survival, which cannot be evaluated on a cohort of 1082 patients, the precision of their score based on a strictly theoretical analysis of probability.

Response 2:

Although TNM staging is clinically important for the development of diagnostic and therapeutic strategies related to prognosis, it does not yet fully meet the clinical needs in clinical application[8,9]. This study is a preliminary attempt based on a single-center database. In the future, we will try multi-center and prospective studies with your comments to further validate the clinical value of this model.

Comments 3: Finally, they validate their method by a new statistical analysis using a concordance score (C index) to validate their IDI and NRI method. They then allow themselves to conclude that their score has better predictive power than the TNM score... which is not the point!

Response 3:

Concordance index (C-index) is an index that can be used to judge the discrimination ability of various models, and can evaluate the accuracy of model prediction results.Net reclassification improvement (NRI) and integrated discrimination improvement (IDI) are two complementary validation methods. NRI is primarily used to compare the predictive ability of the old model with the new model, assessing improvement only when a specific cut-off point is established. IDI is mainly used to test the overall improvement of the model and evaluate its overall performance. We analyzed and compared the predictive performance of nomogram and conventional TNM staging by using C-index, NRI, and IDI. The results showed that the prediction accuracy of the nomogram was significantly better than that of the traditional TNM staging. This analysis method has also been supported and confirmed in previous studies[10-13].

Comments 4: Furthermore, a non-negligible bias of this type of study is not to take into account the impossibility of defining the importance of each factor T, N or M which could vary from one to the other at each stage, and that ACE has never been defined as a correlating factor for these stages. Each stage T, N or M should be analyzed independently, which is not possible with a cohort of 1000 patients.

Response 4:

Thank you for your valuable suggestions. As you mentioned, we regret indeed that we are limited by the sample size and are not able to conduct the more refined analyses mentioned above at this stage. In the future, we will try to address this shortcoming through further multi-centre and longitudinal studies.

Comments 5: Finally, the clinical interest of this nomogram score is not analyzed. Indeed, it does not specify the clinical benefit of the nomogram. It does not specify whether it should have an impact on therapeutic care. It has no real interest in patient care, nor in the statistical reclassification of cancers, the subject of the TNM score.

Response 5:

Thank you for your guidance and criticism of the study. The aim of this study was to predict the prognostic overall survival by constructing a nomogram for stage I-III CRC patients with normal preoperative CEA. By personalizing the scores for different patients, we can screen out the high-risk group that may have a poor prognosis in advance, so that we can intervene in advance to prolong the survival of the patients. In addition, in this study, we constructed the model through the training set and applied it to the validation set. The results showed that the model has good discriminative ability and prediction effect in the training set and the validation set. This suggests that the model has potential clinical applications. However, the model's complementarity to clinical diagnosis and treatment strategies as well as reclassification methods needs to be further clarified by subsequent prospective clinical trials, and we will strive to validate the model's predictive ability through multi-center studies.

3. Response to Comments on the Quality of English Language

Point 1: Difficult to understand

Response 1:

Thanks for your suggestion. We feel sorry for our poor writing. To improve the quality of language in our article, the manuscript has been revised and edited by English language services of MDPI ( The proof can be found below). We hope the revised manuscript could be acceptable to you.

4. Additional clarifications

[I] We described in detail the specific research steps of statistical analysis. In addition, the statistical analysis methods used for continuous variables and categorical variables have been supplemented in Table 1, supplemental Table 2 and so on. Hope to get your approval.

[II] To improve the quality of language in our articles, the manuscript has been revised and edited by the English language services of MDPI ( The proof can be found below).

[III] We added a new table about the R package used in this study(supplemental table S1,can be found below) and the original supplemental table S1 is named supplemental table S2.

  1. Miller, K.D.; Nogueira, L.; Devasia, T.; Mariotto, A.B.; Yabroff, K.R.; Jemal, A.; Kramer, J.; Siegel, R.L. Cancer treatment and survivorship statistics, 2022. CA Cancer J Clin 2022, 72, 409-436, doi:10.3322/caac.21731.
  2. Cheng, E.; Blackburn, H.N.; Ng, K.; Spiegelman, D.; Irwin, M.L.; Ma, X.; Gross, C.P.; Tabung, F.K.; Giovannucci, E.L.; Kunz, P.L.; et al. Analysis of Survival Among Adults With Early-Onset Colorectal Cancer in the National Cancer Database. JAMA network open 2021, 4, e2112539, doi:10.1001/jamanetworkopen.2021.12539.
  3. Konishi, T.; Shimada, Y.; Hsu, M.; Tufts, L.; Jimenez-Rodriguez, R.; Cercek, A.; Yaeger, R.; Saltz, L.; Smith, J.J.; Nash, G.M.; et al. Association of Preoperative and Postoperative Serum Carcinoembryonic Antigen and Colon Cancer Outcome. JAMA Oncol 2018, 4, 309-315, doi:10.1001/jamaoncol.2017.4420.
  4. Chansky, K.; Detterbeck, F.C.; Nicholson, A.G.; Rusch, V.W.; Vallières, E.; Groome, P.; Kennedy, C.; Krasnik, M.; Peake, M.; Shemanski, L.; et al. The IASLC Lung Cancer Staging Project: External Validation of the Revision of the TNM Stage Groupings in the Eighth Edition of the TNM Classification of Lung Cancer. Journal of thoracic oncology : official publication of the International Association for the Study of Lung Cancer 2017, 12, 1109-1121, doi:10.1016/j.jtho.2017.04.011.
  5. Ekeblad, S.; Skogseid, B.; Dunder, K.; Oberg, K.; Eriksson, B. Prognostic factors and survival in 324 patients with pancreatic endocrine tumor treated at a single institution. Clinical cancer research : an official journal of the American Association for Cancer Research 2008, 14, 7798-7803, doi:10.1158/1078-0432.Ccr-08-0734.
  6. Msika, S.; Benhamiche, A.M.; Jouve, J.L.; Rat, P.; Faivre, J. Prognostic factors after curative resection for gastric cancer. A population-based study. European journal of cancer (Oxford, England : 1990) 2000, 36, 390-396, doi:10.1016/s0959-8049(99)00308-1.
  7. Liang, W.; Zhang, L.; Jiang, G.; Wang, Q.; Liu, L.; Liu, D.; Wang, Z.; Zhu, Z.; Deng, Q.; Xiong, X.; et al. Development and validation of a nomogram for predicting survival in patients with resected non-small-cell lung cancer. Journal of clinical oncology : official journal of the American Society of Clinical Oncology 2015, 33, 861-869, doi:10.1200/jco.2014.56.6661.
  8. Weiser, M.R.; Landmann, R.G.; Kattan, M.W.; Gonen, M.; Shia, J.; Chou, J.; Paty, P.B.; Guillem, J.G.; Temple, L.K.; Schrag, D.; et al. Individualized prediction of colon cancer recurrence using a nomogram. Journal of clinical oncology : official journal of the American Society of Clinical Oncology 2008, 26, 380-385, doi:10.1200/jco.2007.14.1291.
  9. Amin, M.B.; Greene, F.L.; Edge, S.B.; Compton, C.C.; Gershenwald, J.E.; Brookland, R.K.; Meyer, L.; Gress, D.M.; Byrd, D.R.; Winchester, D.P. The Eighth Edition AJCC Cancer Staging Manual: Continuing to build a bridge from a population-based to a more "personalized" approach to cancer staging. CA Cancer J Clin 2017, 67, 93-99, doi:10.3322/caac.21388.
  10. Wu, J.; Zhang, H.; Li, L.; Hu, M.; Chen, L.; Xu, B.; Song, Q. A nomogram for predicting overall survival in patients with low-grade endometrial stromal sarcoma: A population-based analysis. Cancer communications (London, England) 2020, 40, 301-312, doi:10.1002/cac2.12067.
  11. Cui, Y.; Zhang, J.; Li, Z.; Wei, K.; Lei, Y.; Ren, J.; Wu, L.; Shi, Z.; Meng, X.; Yang, X.; et al. A CT-based deep learning radiomics nomogram for predicting the response to neoadjuvant chemotherapy in patients with locally advanced gastric cancer: A multicenter cohort study. EClinicalMedicine 2022, 46, 101348, doi:10.1016/j.eclinm.2022.101348.
  12. Miao, W.; Nie, P.; Yang, G.; Wang, Y.; Yan, L.; Zhao, Y.; Yu, T.; Yu, M.; Wu, F.; Rao, W.; et al. An FDG PET/CT metabolic parameter-based nomogram for predicting the early recurrence of hepatocellular carcinoma after liver transplantation. European journal of nuclear medicine and molecular imaging 2021, 48, 3656-3665, doi:10.1007/s00259-021-05328-w.
  13. Deng, X.; Hou, H.; Wang, X.; Li, Q.; Li, X.; Yang, Z.; Wu, H. Development and validation of a nomogram to better predict hypertension based on a 10-year retrospective cohort study in China. Elife 2021, 10, doi:10.7554/eLife.66419.

English-Editing-Certificate-71868

Supplemental TableS1. The R packages used in this study.

Function

Packages

Random grouping(training set and validation set)

“caret” package

Predictors selection

“survival” and “MASS” packages

Nomogram and calibration curves

“rms” package

Receiver operating characteristic curves

“survivalROC” package

Decision curve analysis

“dcurves” package

Net reclassification improvement

“nricens” package

Integrated discrimination improvement

“survIDINRI” package

C-index and C-index change

“CsChange” package

Kaplan-Meier curves and Risk stratification

“survival”package

Reviewer 2 Report

Comments and Suggestions for Authors

The article describes a extensively tested nomogram with excellent correlation with survival. Although I would like to see the effect on clinical practice/desicion making in patients I know it is to early for that. I would encourage the authors to further investigate this and evaluate the model in multicenter studies. But all in all very good work that warrants publication.

Author Response

1. Summary

Thank you for considering our article for publication. We are pleased to hear that you found our nomogram to have an excellent correlation with survival. We will further explore the predictive power of this model through a multi-center study. Your insights are invaluable to us, and we are committed to improving our work based on your suggestions.  In addition, we have resubmitted a new manuscript in the revised state, with the revisions highlighted in red. Once again, we sincerely appreciate your positive feedback and your encouragement to pursue further research.

(Revised manuscript can be found in attachment “revised manuscript”and revised manuscript-supplementary can be found in attachment “revised manuscript-supplementary”)

2. Point-by-point response to Comments and Suggestions for Authors

Comments 1: Although l would like to see the effect on clinical practice/decision making in patients I know it is too early for that. I would encourage the authors to further investigate this and evaluate the model in multicenter studies.

Response 1:

We sincerely thank you for your high recognition of our work, which is a great support and encouragement for our research. This study is a retrospective study based on a real-world population. We constructed and validated a clinical prediction model for patients with normal preoperative CEA in stages I-III. The results showed that the model had superior predictive ability and clinical applicability, and was significantly better than TNM staging. Indeed, limited by a single database, the predictive value of the model is not yet well demonstrated. We will further investigate the clinical value of this model through multi-center studies. Finally, thank you again for your support!

3. Response to Comments on the Quality of English Language

Point 1: English language fine. No issues detected

Response 1:

Thank you very much for your recognition! To further improve the quality of language in our articles, the manuscript has been revised and edited by the English language services of MDPI.

4. Additional clarifications

[I] We described in detail the specific research steps of statistical analysis. In addition, the statistical analysis methods used for continuous variables and categorical variables have been supplemented in table 1, supplemental table S2 and so on. Hope to get your approval!

[II] To improve the quality of language in our articles, the manuscript has been revised and edited by the English language services of MDPI ( The proof can be found below).

[III] We added a new table about the R package used in this study(supplemental table S1,can be found below) and the original supplemental table S1 is named supplemental table S2.

Supplemental TableS1. The R packages used in this study.

Function

Packages

Random grouping(training set and validation set)

“caret” package

Predictors selection

“survival” and “MASS” packages

Nomogram and calibration curves

“rms” package

Receiver operating characteristic curves

“survivalROC” package

Decision curve analysis

“dcurves” package

Net reclassification improvement

“nricens” package

Integrated discrimination improvement

“survIDINRI” package

C-index and C-index change

“CsChange” package

Kaplan-Meier curves and Risk stratification

“survival”package

Reviewer 3 Report

Comments and Suggestions for Authors

This is an interesting study and the study design seems well described. 

However, there are some issues that needs clarification:

The authors have measured a broad spectrum of tumor markers. They end up selecting two tumormarkes in their model. The tumormarkers were selected for the nomogram based on a multivariable analysis. The authors must describe why those two markers where chosen because in the univariable analysis other markers have shown statitical significance as well.  Did the authors take choose markers with the lowest p-values??? This must be adressed in the manuscript. In my opinion this is very critical and a major limitation in this study. 

Further the authors are encouraged to discuss if the different follow-up regimens (CT, MRI, chest CT ) blood test etc., could affect the result.

The limitations of the study must be described in more details in the discussion section. Especially that the validation is done on data on patients that are representative of all the patients included. 

They must mention the total number of patients in the database (top box in figure 1) so the readers of the article can judge if there are any selection bias. I guess that the total number is 1359 +1492. 

Author Response

1. Summary

Thank you very much for your comments and professional advice. These opinions help to improve the academic rigor of our article. Based on your suggestion and request, we have made corrected modifications on the revised manuscript. Meanwhile, The manuscript had be revised and edited by English language services of MDPI. In addition, we have resubmitted a new manuscript in the revised state, with the revisions highlighted in red..The main corrections in the paper and the responds to your comments are as following:

(Revised manuscript can be found in attachment “revised manuscript”and revised manuscript-supplementary can be found in attachment “revised manuscript-supplementary”)

2. Point-by-point response to Comments and Suggestions for Authors

Comments 1: The authors have measured a broad spectrum of tumor markers. They end up selecting two tumor markers in their model. The tumor markers were selected for the nomogram based on a multivariable analysis. The authors must describe why those two markers were chosen because in the univariable analysis, other markers have shown statistical significance as well. 

Response 1:

Thank you for your valuable comments. Allow me to explain the selection criteria for serum tumour markers. This study was a retrospective analysis conducted in a real-world setting, where serum tumour marker levels were based on actual patients. Although univariate Cox proportional risk regression analysis showed a variety of serum tumour markers to be statistically significant in predicting survival, however, it does not effectively exclude the influence of confounding factors on clinical outcomes and cannot effectively predict outcomes. In contrast, multivariate Cox regression analysis is more effective in taking control of confounding factors and thus more accurately discussing the effect of the independent variable on the dependent variable. Therefore, it is reasonable to believe that serum tumour markers (CA242, CA125) with statistical significance in multivariate analysis are independent factors for patients undergoing radical resection of colorectal cancer with normal preoperative CEA (P < 0.05).

Comments 2: Did the authors take choose markers with the lowest p-values??? This must be addressed in the manuscript.

Response 2:

Allow me to elaborate on the statistical analysis of the study. We selected prognostic factors for OS by Cox proportional hazards regression analysis. Specifically, we first included all variables for univariate Cox regression analysis, followed by multivariate Cox regression analysis for variables that were statistically significant in the univariate analysis (P < 0.05). The results of multifactorial regression showed age, Histologic type, T stage, N stage, carbohydrate antigen 242 (CA242), and carbohydrate antigen 125 (CA125) were independent prognostic factors for OS (p < 0.05). In this study, a p value <0.05 was considered statistically significant. We appreciate your questions. In order to provide readers with a clearer understanding of the research methodology, we further refine the statistical analysis steps(line 133-135).

Comments 3: Further the authors are encouraged to discuss if the different follow-up regimens (CT, MRI, chest CT ) blood test etc., could affect the result.

Response 3:

Due to the limitation of hospital conditions and habits, most of our patients adopt the follow-up method of path, so the analysis can not be carried out at present, but as the number of cases accumulates further, we will try to conduct the relevant analyses according to the recommendations to determine whether the different means and methods of follow-up have an impact on the results of the follow-ups.

Comments 4: The limitations of the study must be described in more details in the discussion section. Especially that the validation is done on data on patients that are representative of all the patients included. 

Response 4:

We appreciate your valuable comments. Due to the lack of an external database, we were only able to conduct research based on a single database. We collected the clinical data from more than 1,000 patients with stage I-III CRC who underwent radical resection and had normal preoperative CEA in the Affiliated Hospital of Shanghai Jiaotong University School of Medicine in China, and randomly divided these patients into a training set (70%) and a validation set (30%), and then constructed a model based on the training set and validated it based on the validation set.The limitations of the study are described in more details in the discussion section.(See line 355-365 for details)

Comments 5: They must mention the total number of patients in the database (top box in figure 1) so the readers of the article can judge if there are any selection bias. I guess that the total number is 1359 +1492. 

Response 5:

We sincerely appreciate your advice. Your guess is correct. The total number of patients in our database is 2851. We have added the total number of patients in our database to the flowchart(Figure 1).

3. Response to Comments on the Quality of English Language

Point 1: English lanquage fine. No issues detected.

Response 1:

Thank you very much for your recognition! To further improve the quality of language in our articles, the manuscript has been revised and edited by the English language services of MDPI.

4. Additional clarifications

[I] We described in detail the specific research steps of statistical analysis. In addition, the statistical analysis methods used for continuous variables and categorical variables have been supplemented in table 1, supplemental table S2 and so on. Hope to get your approval!

[II] To improve the quality of language in our articles, the manuscript has been revised and edited by the English language services of MDPI ( The proof can be found below).

[III] We added a new table about the R package used in this study(supplemental table S1,can be found below) and the original supplemental table S1 is named supplemental table S2.

Supplemental TableS1. The R packages used in this study.

Function

Packages

Random grouping(training set and validation set)

“caret” package

Predictors selection

“survival” and “MASS” packages

Nomogram and calibration curves

“rms” package

Receiver operating characteristic curves

“survivalROC” package

Decision curve analysis

“dcurves” package

Net reclassification improvement

“nricens” package

Integrated discrimination improvement

“survIDINRI” package

C-index and C-index change

“CsChange” package

Kaplan-Meier curves and Risk stratification

“survival”package

Reviewer 4 Report

Comments and Suggestions for Authors

From a biostats and clinical epidemiology, this manuscript has been very well planned, executed and reported. Some suggestions for the Authors:

- it would be very intestering to report the prevalence of normal preoperative CEA at CRC staging, for different geographic areas

- line 104, strictly speaking, all patients were assigned to training/test cohorts, more than randomized

- the study flowchart is extremely informative, compliments!

- line 106, better to say "adjuvant chemotherapy"; moreover, details about adjuvant chemotherapy and immunotherapy regimens are totally lacking

- line 116, the median follow-up for the whole cohort is lacking (for surviving patients only)

- line 118, no abdominal ecography scheduled in the standard follow-up?

- line 124, add that continuous covariates are reported as median/IQR (and do that all around the manuscript)

- line 138, please report type/release/packages of the stats software (I see it's R)

- table 1, please report T1 and T2 separately; moreover, age as a continuous value

- table 1, please add the number of OS and PFS events, this is a critical info, to be evaluated in relationship with the median follow-up

- line 147 and everywhere, please always report exact p-values with 3-sign digits

- line 174, univariate analyses are to be labelled as Cox proportional hazard regression models

- line 184 better to say "independent determinants" or "independent risk factors"

- table 2, I would have been expecting a negative prognostic role for rectum vs colon cancer, what do you think about?

- figure 2, IMHO the nomogram is able to predict OS (binary event, dead/alive) more than time to OS (time-to-event), please clarify this topic

- figure 3, the performance drawdown in validation cohort need to be carefully underlined

- discussion, I do believe that more comments on already used nomograms could be of huge help for the reader

Comments on the Quality of English Language

minor

Author Response

1. Summary

Thank you for your letter and the constructive comments on this article in your busy schedule. All of us authors have carefully read the comments that you have given us, and have discussed and revised each of these issues.The following is my list of revisions. In addition, we have resubmitted a new manuscript in the revised state, with the revisions highlighted in red. If there are any incorrect answers or questions in the manuscript please do not hesitate to let us know.

(Revised manuscript can be found in attachment “revised manuscript”and revised manuscript-supplementary can be found in attachment “revised manuscript-supplementary”)

2. Point-by-point response to Comments and Suggestions for Authors

Comments 1: it would be very interesting to report the prevalence of normal preoperative CEA at CRC staging, for different geographic areas.

Response 1:

Thank you for your valuable suggestions. Further reporting on the prevalence of patients with normal preoperative CEA in different regions will help readers understand the background of the article more deeply. We have clarified this part further in the discussion (lines 270-275). Hope to get your approval.

Comments 2: strictly speaking, all patients were assigned to training/test cohorts, more than randomized

Response 2:

In this study, we randomly divided the study population into a training set and a validation set in R (“caret” package)for Windows (version 4.2.1, http://www.R-project.org). Randomization allowed the confounders to be balanced across the groups, and baseline conditions were well-comparable between groups, thus allowing the statistically significant factors to be shown.

Comments 3: line 106, better to say "adjuvant chemotherapy"; moreover, details about adjuvant chemotherapy and immunotherapy regimens are totally lacking

Response 3:

Thanks to your suggestion, we have changed "chemotherapy" to "adjuvant chemotherapy"(line 110-111). In this retrospective study, the postoperative adjuvant chemotherapy regimens used were all first-line adjuvant chemotherapy regimens (FOLFOX or XELOX regimens). However, considering that immunotherapy is not a first-line treatment option, it was not addressed in this study. In addition, we all followed the latest NCCN guidelines for postoperative adjuvant chemotherapy for postoperative pathology indicated stage III as well as some stage II patients with high-risk factors.

Comments 4: line 116, the median follow-up for the whole cohort is lacking (for surviving patients only)

Response 4:

Thanks to your valuable suggestions, our median follow-up has been added to Table1.

Comments 5: line 118, no abdominal echography scheduled in the standard follow-up?

Response 5:

We appreciate your inquiry. In our hospital, we usually use enhanced CT or enhanced MRI and rarely ultrasonography in the abdominal follow-up of CRC patients. In recent years, with the gradual maturation of ultrasonography in our hospital, we have begun to use ultrasonography for the follow-up of abdominal conditions in some of our patients who are allergic to contrast media or have other contraindications. Therefore, abdominal ultrasonography is not yet widely used in follow-up planning.

Comments 6: line 124, add that continuous covariates are reported as median/IQR (and do that all around the manuscript)

Response 6:

Continuous covariates have been added to Table 1, and hope to get your approval..

Comments 7: line 138, please report type/release/packages of the stats software (I see it's R)

Response 7:

Thank you for your advice. We have added details of the R software for Windows (version 4.2.1, http://www.R-project.org) and made tables of the R software packages used in this study (Supplementary Table 1).

Comments 8:table 1, please report T1 and T2 separately; moreover, age as a continuous value

Response 8:

We have further added in Table 1. Please see Table 1 for more specific information.

Comments 9:table 1, please add the number of OS and PFS events, this is a critical info, to be evaluated in relationship with the median follow-up

Response 9:

Since the population we studied was stage I-III CRC patients, I assume you're talking about DFS and not PFS. We have further added them in Table 1. Please see Table 1 for more specific information.

Comments 10:line 147 and everywhere, please always report exact p-values with 3-sign digits

Response 10:

We thank you for your careful reading. We have checked the full text to ensure that all p-values are 3-digit valid numbers. Changes have been highlighted in red text.

Comments 11: line 174, univariate analyses are to be labelled as Cox proportional hazard regression models

Response 11:

We apologize for the oversight in our work. We have replaced "univariate analyses" with "univariate Cox proportional hazard regression".(line 193)

Comments 12:line 184 better to say "independent determinants" or "independent risk factors"

Response 12:

Thank you for your valuable comments. We have made the replacement.(line 203-204)

Comments 13: table 2, I would have been expecting a negative prognostic role for rectum vs colon cancer, what do you think about?

Response 13:

This study was conducted at the Hospital Affiliated to Shanghai Jiaotong University School of Medicine, Shanghai, China. Given regional differences and different patient populations, this does not reflect the prognosis of all patients. This issue needs to be further studied through multicenter studies.

Comments 14: figure 2, IMHO the nomogram is able to predict OS (binary event, dead/alive) more than time to OS (time-to-event), please clarify this topic

Response 14:

In this retrospective study, we assessed the prognostic significance using univariate and multivariate Cox proportional hazards regression analyses that were able to take time into account. The clinical prediction model was based on a multivariate Cox hazard risk regression model, so the nomogram also took time and time to outcome into account. In other studies[1-3] that constructed nomograms based on the Cox regression model, the time variable was similarly taken into account.

Comments 15: figure 3, the performance drawdown in validation cohort need to be carefully underlined

Response 15:

We think this is a good suggestion. The C-index of the validation set was 0.702, which can be used to predict the overall survival of CRC patients with normal preoperative CEA. However, this value of the validation set is smaller than the C-index of the training set (0.748). This suggests a decrease in the predictive performance of the validation set, which we believe may be due to the inconsistent baselines of the two data sets. To give the reader a more complete understanding of the article, we further elaborated on the limitations at the end of the article (last paragraph of the discussion,line 357-361).

Comments 16:discussion, I do believe that more comments on already used nomograms could be of huge help for the reader

Response 16:

Thank you for your advice. In order to give readers a deeper understanding of the clinical value of nomograms, we have added some prognostic nomograms combined with serum tumour markers. The revised part has been added to the discussion, and I hope it will be approved by you.(line 322-329)

3. Response to Comments on the Quality of English Language

Point 1: Minor editing of English language required

Response 1:

Thank you very much for your suggestion! To further improve the quality of language in our articles, the manuscript has been revised and edited by the English language services of MDPI ( The proof can be found below)..

4. Additional clarifications

[I] We described in detail the specific research steps of statistical analysis. In addition, the statistical analysis methods used for continuous variables and categorical variables have been supplemented in table 1, supplemental table S2 and so on. Hope to get your approval!

[II] To improve the quality of language in our articles, the manuscript has been revised and edited by the English language services of MDPI  ( The proof can be found below).

[III] We added a new table about the R package used in this study(supplemental table S1,can be found below) and the original supplemental table S1 is named supplemental table S2.

  1. Wu, J.; Zhang, H.; Li, L.; Hu, M.; Chen, L.; Xu, B.; Song, Q. A nomogram for predicting overall survival in patients with low-grade endometrial stromal sarcoma: A population-based analysis. Cancer communications (London, England) 2020, 40, 301-312, doi:10.1002/cac2.12067.
  2. Liang, W.; Zhang, L.; Jiang, G.; Wang, Q.; Liu, L.; Liu, D.; Wang, Z.; Zhu, Z.; Deng, Q.; Xiong, X.; et al. Development and validation of a nomogram for predicting survival in patients with resected non-small-cell lung cancer. Journal of clinical oncology : official journal of the American Society of Clinical Oncology 2015, 33, 861-869, doi:10.1200/jco.2014.56.6661.
  3. Lin, J.X.; Wang, Z.K.; Hong, Q.Q.; Zhang, P.; Zhang, Z.Z.; He, L.; Wang, Q.; Shang, L.; Wang, L.J.; Sun, Y.F.; et al. Assessment of Clinicopathological Characteristics and Development of an Individualized Prognostic Model for Patients With Hepatoid Adenocarcinoma of the Stomach. JAMA network open 2021, 4, e2128217, doi:10.1001/jamanetworkopen.2021.28217.

Supplemental TableS1. The R packages used in this study.

Function

Packages

Random grouping(training set and validation set)

“caret” package

Predictors selection

“survival” and “MASS” packages

Nomogram and calibration curves

“rms” package

Receiver operating characteristic curves

“survivalROC” package

Decision curve analysis

“dcurves” package

Net reclassification improvement

“nricens” package

Integrated discrimination improvement

“survIDINRI” package

C-index and C-index change

“CsChange” package

Kaplan-Meier curves and Risk stratification

“survival”package